# Projection of Sediment Loading from Pearl River Basin, Mississippi into Gulf of Mexico under a Future Climate with Afforestation

Ying Ouyang [1,*] , Yanbo Huang [2] , Prem B. Parajuli [3] , Yongshan Wan [4] , Johnny M. Grace [5] , Peter V. Caldwell [6] and Carl Trettin [7]

1   USDA Forest Service, Center for Bottomland Hardwoods Research, Southern Research Station, 775 Stone Blvd., Thompson Hall, Room 309, Mississippi State, MS 39762, USA
2   Genetics and Sustainable Agriculture Research Unit, Crop Science Research Laboratory, USDA-Agricultural Research Service, 810 Highway 12 East, Mississippi State, MS 39762, USA; yanbo.huang@usda.gov
3   Department of Agricultural and Biological Engineering, Mississippi State University, Mississippi State, MS 39762, USA; pparajuli@abe.msstate.edu
4   Center for Environmental Measurement and Modeling, US EPA, 1 Sabine Island Drive, Gulf Breeze, FL 32561, USA; wan.yongshan@epa.gov
5   USDA Forest Service, Center for Forest Watershed Research, Southern Research Station, 1740 S. Martin Luther King Jr. Blvd., Perry-Paige Bldg., Suite 303 North, Tallahassee, FL 32307, USA; johnny.m.grace@usda.gov
6   USDA Forest Service, Center for Integrated Forest Science, Southern Research Station, 3160 Coweeta Lab Road, Otto, NC 28763, USA; peter.v.caldwell@usda.gov
7   USDA Forest Service enter for Forest Watershed Research, 3734 Hwy 402, Cordesville, SC 29434, USA; carl.c.trettin@usda.gov
*   Correspondence: ying.ouyang@usda.gov

**Abstract:** Sediment load in rivers is recognized as both a carrier and a potential source of contaminants. Sediment deposition significantly changes river flow and morphology, thereby affecting stream hydrology and aquatic life. We projected sediment load from the Pearl River basin (PRB), Mississippi into the northern Gulf of Mexico under a future climate with afforestation using the SWAT (Soil and Water Assessment Tool)-based HAWQS (Hydrologic and Water Quality System) model. Three simulation scenarios were developed in this study: (1) the past scenario for estimating the 40-year sediment load from 1981 to 2020; (2) the future scenario for projecting the 40-year sediment load from 2025 to 2064, and (3) the future afforestation scenario that was the same as the future scenario, except for converting the rangeland located in the middle section of the Pearl River watershed of the PRB into the mixed forest land cover. Simulations showed a 16% decrease in sediment load for the future scenario in comparison to the past scenario due to the decrease in future surface runoff. Over both the past and future 40 years, the monthly maximum and minimum sediment loads occurred, respectively, in April and August; whereas the seasonal sediment load followed the order: spring > winter > summer > fall. Among the four seasons, winter and spring accounted for about 86% of sediment load for both scenarios. Under the future 40-year climate conditions, a 10% reduction in annual average sediment load with afforestation was observed in comparison to without afforestation. This study provides new insights into how a future climate with afforestation would affect sediment load into the northern Gulf of Mexico.

**Keywords:** afforestation; climate change; HAWQS model; Gulf of Mexico; sediment load

## 1. Introduction

Sediment load in surface waters is a potential source of contaminants to aquatic environments as sediment adsorbs and transports toxic chemicals, excess nutrients, and pathogens, while sediment deposition can change stream flow and morphology and thereby has broad impacts on hydrologic channels, aquatic life, and recreation activity [1]. Agricultural practices, forest disturbances, and urbanization activities are the major sources of

sediment contamination and deposition in rivers, streams, and lakes [2–5]. When the rates and quantities of soil erosion and sediment load are sufficient to adversely affect terrestrial communities and streams, the surface water resources are impaired by sediments [6]. Consequently, this is occurring in the lower Mississippi River basin (LMRB) which discharges to the northern Gulf of Mexico (NGOM). This basin is one of the most disturbed by human activity among the world's largest coastal and river basins [5,7]. From the 1950s to the 1970s, the LMRB underwent a widespread loss of bottomland hardwood forests due to the clearcuttings for crop production, flood control, and floodplain development [8]. Such anthropogenic activities are largely responsible for the increased sediment loads in the surface water systems of the LMRB and NGOM [9]. Milliman and Meade [10] reported that the Mississippi River delivered 210 Mt (million tons)/y of sediment to the NGOM from the 1950s to the 1980s, while Bentley et al. [11] estimated that the lower reaches of the Mississippi and Atchafalaya Rivers discharged 57 and 71.5 Mt/y of sediment, respectively, from Tarbert Landing and Simmesport into the NGOM from 2008 to 2010. Xu et al. [12] reviewed and synthesized sediment dynamics in the Mississippi River deltaic plain. These authors suggested that future efforts should be focused on enhancing river sediment delivery, increasing sediment retention, and minimizing soil erosion.

Despite numerous studies having been made to investigate the sediment load from the LMRB into the NGOM [5,13–17], limited efforts have been devoted to estimating the impacts of forest watersheds, and their restoration practices, on sediment load in the region. Ouyang et al. [5] applied the HSPF (Hydrological Simulation Program-FORTRAN) model to predict the role of afforestation on sediment load in the lower Yazoo River Watershed, Mississippi. They reported that a conversion of marginal agricultural land into forests reduced sediment load. In general, a two-fold increase in forest land area resulted in approximately a two-fold reduction in annual sediment load, which occurred because forests absorb water, reduce surface water runoff, and prevent soil erosion. However, a thorough literature review reveals that very few efforts have been undertaken to estimate the contributions of forest watersheds to sediment delivery from the LMRB into the NGOM under a changing climate. This knowledge is critical to the assessment of relations among afforestation, climatic, environmental, and economic impacts in the region.

The goal of this study was to predict sediment load from the Pearl River basin (PRB), Mississippi (within the LMRB) into the NGOM under the past climate and the future climate with afforestation, using the Hydrologic and Water Quality System (HAWQS) model. Our specific objectives were to: (1) create and calibrate the PRB model; and (2) apply the model to estimate sediment load from the PRB into the NGOM over the past 40 years (1981 to 2020), future 40 years (2025 to 2064), and future 40 years with afforestation (i.e., conversion of a rangeland into a mixed forest land).

## 2. Materials and Methods

### 2.1. Study Site

The PRB covers southeastern Louisiana and southwestern Mississippi (Figure 1) with a drainage area of 22,533 km², encompassing all or parts of the 24 counties in Mississippi and 3 parishes in Louisiana. The Pearl River in the basin originates from the east central region of Mississippi and flows into the NGOM with a length of about 790 km. Nearly one million people live in the basin and more than one-third are residents of Mississippi [18]. The basin consists of five watersheds, namely the upper Pearl River (Hydrologic Unit Code (HUC) 03180001, 6379 km²), middle Pearl River (HUC 03180002, 5120 km²), lower-middle Pear River (HUC 03180003, 3156 km²), lower Pearl River (HUC 03180004, 4717 km²), and Bogue Chitto River (HUC 03180005, 3130 km²) watersheds. Average maximum temperature is 33.4 °C in July and average minimum temperature is 0.3 °C in January. Annual average precipitation is 1412 mm with a wet period from November to April and a dry period from May to October. The PRB is on the 303(d) list of impaired waterways due to excessive nutrient levels (nitrogen and phosphorus), mercury concentrations, sediment loadings, and pesticide concentrations [19]. The basin is dominated by 69% forest, followed by 27%

agricultural land, and 3% wetland and water area [20]. The predominant soil types in the basin are fine sandy loam and silt loam soils [21]. The basin is the largest area of intact bottomland hardwood forest remaining in the southeastern US with a total of more than 14,000 ha, which is composed of mixed bottomland hardwood forest dominated by various oaks (*Quercus* spp.), sweetgum (*Liquidambar stryaciflua*), hickories (*Carya* spp.), and elms (*Ulmus* spp.) [22].

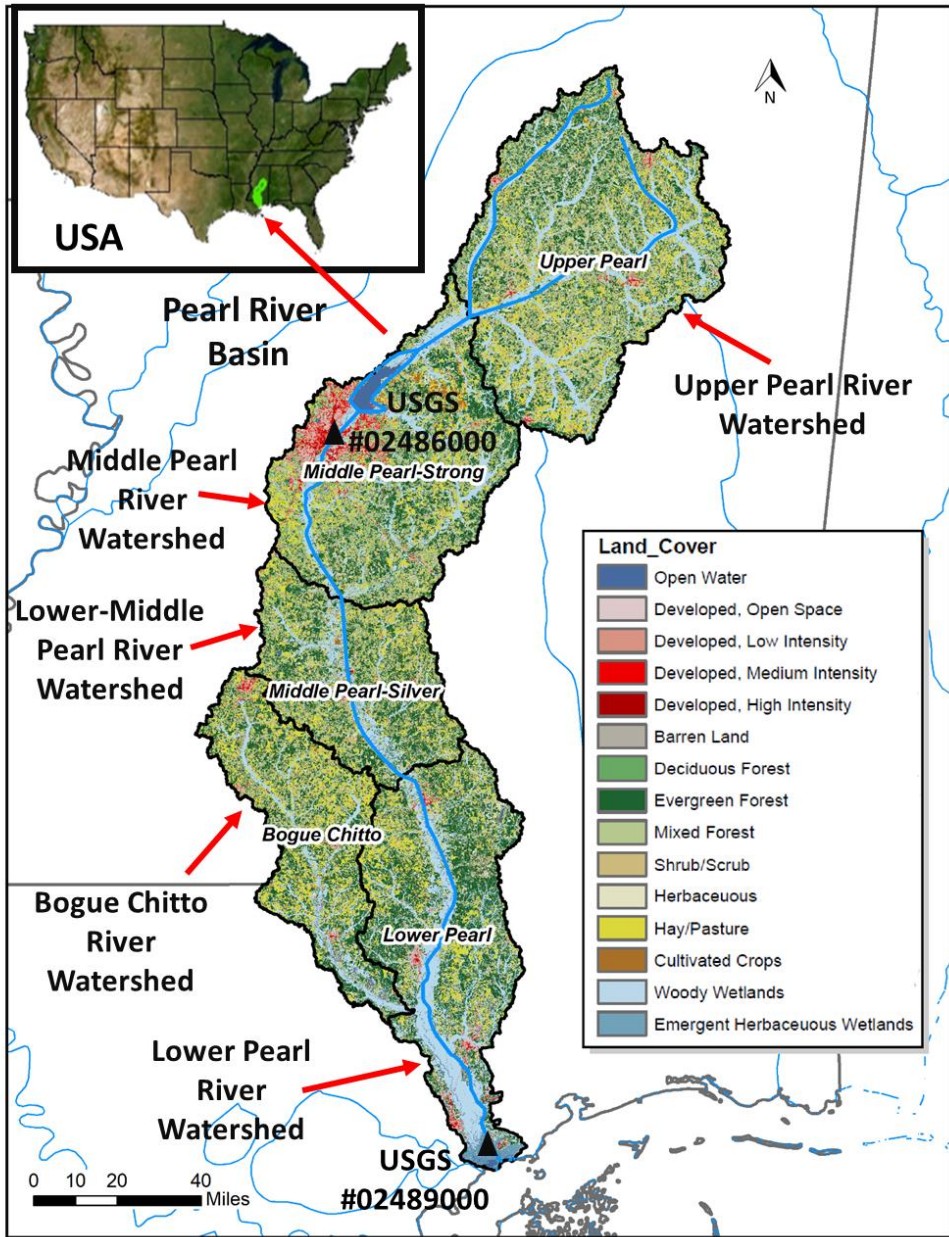

**Figure 1.** Location and land use of the Pear River basin in Mississippi, USA.

## 2.2. HAWQS Model Description

HAWQS is a SWAT (Soil and Water Assessment Tool)-based model with a user-friendly interface and it is used to simulate hydrological processes and water quality in watersheds of continental USA, in conjunction with impacts of land use, land management, and climate change [23]. The past and future weather datasets are pre-loaded into HAWQS for users' convenience. In recent years, several modeling studies with HAWQS have been reported in the literature. Among them, Yen et al. [24] applied HAWQS to predict water quantity and quality in the Illinois River watershed; Fant et al. [25] used HAWQS to simulate future water

quality conditions and economic impact in the US; and Ouyang [26] employed HAWQS to estimate ET in forest lands of Mississippi. Additional applications of HAWQS can be found elsewhere [27–30]. Overall, HAWQS is a timesaving and cost-effective modeling system for simulating water quantity and quality in complex watersheds. However, it should be noted that any limitations inherited by the SWAT model are also applied to the HAWQS model. Additionally, HAWQS has fewer input parameters that can be used for model calibration and is not as flexible as SWAT.

The PRB model was created using HAWQS with the following three major steps: (1) Create a project. The PRB model was created at the data resolution of HUC-8 for the downstream HUC # 03180004 (or lower Pearl River watershed). After entering this HUC number into the map options of HAWQS, the additional four upstream watersheds contributing to the lower Pearl River watershed were included. This created the entire PRB model (Figure 1); (2) Set the HRU (Hydrologic Response Unit). A threshold level of 1% for HRUs was used to eliminate the effects of minor land uses, soils, and slopes in each watershed; and (3) Develop scenarios. Three simulation scenarios were developed in this study: the past scenario for simulating the past 40-year sediment load from 1981 to 2020; the future scenario for projecting the future 40-year sediment load from 2025 to 2064, and the future afforestation scenario for projecting the future 40-year sediment load from 2025 to 2064 associated with conversion of rangeland into mixed forest land in the middle Pearl River watershed of the PRB. It should be noted that the general management input parameter values such as initial leaf area index, number of heat units to bring plant to maturity, and width of edge-of-field filter strip were changed when converting the rangeland to the mixed forest land. The weather data for the past scenario were from PRISM (Parameter-elevation Regressions on Independent Slopes Model), while the weather data for the future scenario were from CCSM4-RCP85 (Community Climate System Model version 4—Representative Concentration Pathway 85). Both datasets are readily available in HAWQS. Table 1 lists the major modeling input parameter values used in this study. An elaborate description on how to develop and execute the HAWQS model can be found in HAWQS [23]. A flowchart showing the HAWQS modeling procedure is given in Figure 2.

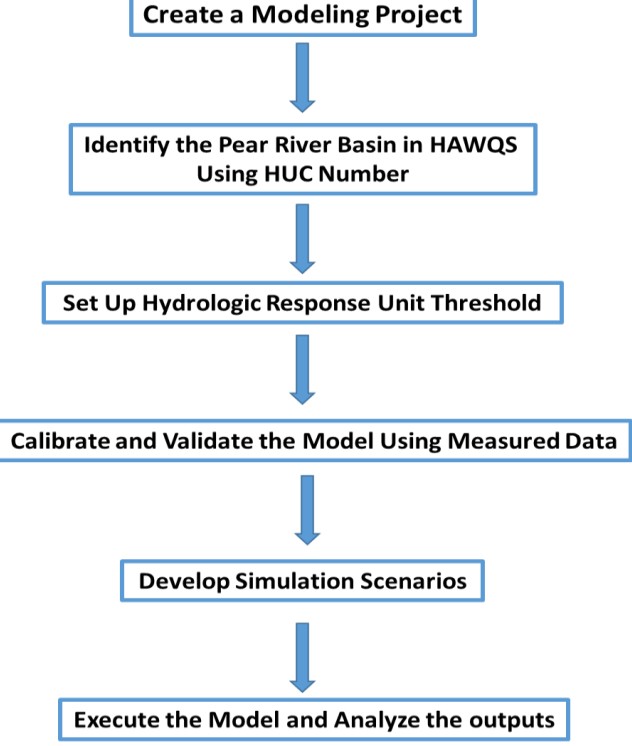

**Figure 2.** The flowchart of HAWQS modeling procedures used in this study.

**Table 1.** Major input parameter values used for the PRB model.

| Parameter | Definition | Value | Unit/method/Explanation | Reference |
|---|---|---|---|---|
| SFTMP | Snowfall temperature | 1 | °C | Local observation |
| SMTMP | Snowmelt base temperature | 0.5 | °C | Local observation |
| SMFMX | Melt factor for snow on June 21 | 4.5 | mm $H_2O$/°C-day | Local observation |
| SMFMN | Melt factor for snow on December 21 | 4.5 | mm $H_2O$/°C-day | Local observation |
| TIMP | TIMP: Snowpack temperature lag factor | 1 | | Local observation |
| IPET | Potential evapotranspiration (PET) method | 2 | Hargreaves method | Calibrated |
| ESCO | Soil evaporation compensation factor | 0.95 | | Calibrated |
| EPCO | EPCO: Plant uptake compensation factor | 1 | | Calibrated |
| ICN | Daily curve number calculation method | 0 | Calculate daily CN value as a function of soil moisture | Calibrated |
| CNCOEF | Plant ET curve number coefficient | 1 | | Calibrated |
| ICRK | Crack flow code | 0 | Do not model crack flow in soil | Local observation |
| SURLAG | Surface runoff lag time | 4 | days | Calibrated |
| CN2 | Subbasins curve number | 0 | | Calibrated |
| IRTE | Channel water routing method | 0 | Variable Storage Method | Calibrated |
| MSK_COL1 | Calibration coefficient used to control impact of the storage time constant for normal flow | 0 | | Calibrated |
| MSK_COL2 | Calibration coefficient used to control impact of the storage time constant for low flow | 3.5 | | Calibrated |
| MSK_X | Weighting factor controlling relative importance of inflow rate and outflow rate in determining water storage in reach segment | 0.2 | | Calibrated |
| TRNSRCH | Fraction of transmission losses from main channel that enter deep aquifer | 0 | | Calibrated |
| EVRCH | Reach evaporation adjustment factor | 1 | | Calibrated |
| IDEG | Channel degradation code | 0 | Channel dimension is not updated as a result of degradation | Local observation |
| PRF | Peak rate adjustment factor for sediment routing in the main channel | 1 | | Calibrated |
| SPCON | Linear parameter for calculating the maximum amount of sediment that can be re-entrained during channel sediment routing | 0 | | Calibrated |
| SPEXP | Exponent parameter for calculating sediment re-entrained in channel sediment routing | 1 | | Calibrated |
| IWQ | In-stream water quality code | 1 | | Calibrated |
| ADJ_PKR | Peak rate adjustment factor for sediment routing in the subbasin (tributary channels) | 0.5 | | Calibrated |
| Weather dataset 1 | PRISM | Time series | Past scenario | Downloaded from HAWQS |
| Weather dataset 2 | CCSM4-RCP85 | Time series | Future scenario | Downloaded from HAWQS |

## 3. Results and Discussion

### 3.1. Model Calibration and Validation

The PRB model was calibrated and validated for the discharges and sediment loads using observed data. The observed discharges were downloaded from the US Geological Survey (USGS) Station #02489000 in the Pearl River near Columbia, MS. The observed sediment data are very limited and were obtained from both stations, #02489000 and #02486000 (near Jackson, MS) (Figure 1). The model calibration was accomplished by adjusting the input parameter values (Table 1) so that the model predictions best matched the field observations, while the model validation was performed to compare the model predictions with another independent set of observations without adjusting any input parameter values. The goodness-of-fit during the model calibration and validation was

determined with coefficient of determination ($R^2$), Nash Sutcliff efficiency (NSE), and percent bias (PBIAS).

Figure 3 compared the predicted and observed discharges during the model calibration for a 10-year period from 1 January 1999 to 31 December 2008 and during the model validation for a 10-year period from 1 January 2009 to 31 December 2018. The values of $R^2$, NSE, and PBIAS for daily discharge were, respectively, 0.62, 0.47, and 9.12 for the calibration (Figure 3a), and 0.66, 0.5, and 19.38 for the validation (Figure 3b). These statistical values suggested that good agreements were attained between the predicted and observed daily discharges during the model calibration and validation [31]. Figure 3c showed the predicted and observed sediment loads during the model calibration from 12 September 1974 to 6 June 1975 as well as during the model validation from 12 March to 28 September 1981. The values of $R^2$, NSE, and PBIAS were, respectively, 0.69, 0.48, and 8.4 for the model calibration (Figure 3c), and 0.91, 0.84, and 119 for the model validation (Figure 3d). Results concluded that reasonable agreements were obtained between the predicted and observed daily sediment loads during the model calibration and validation. It should be noted that the field measured sediment data are very limited, sparse, and intermittent in the PRB. The long-term and continuous sediment data in the PRB are not attainable for rigorous model calibration and validation.

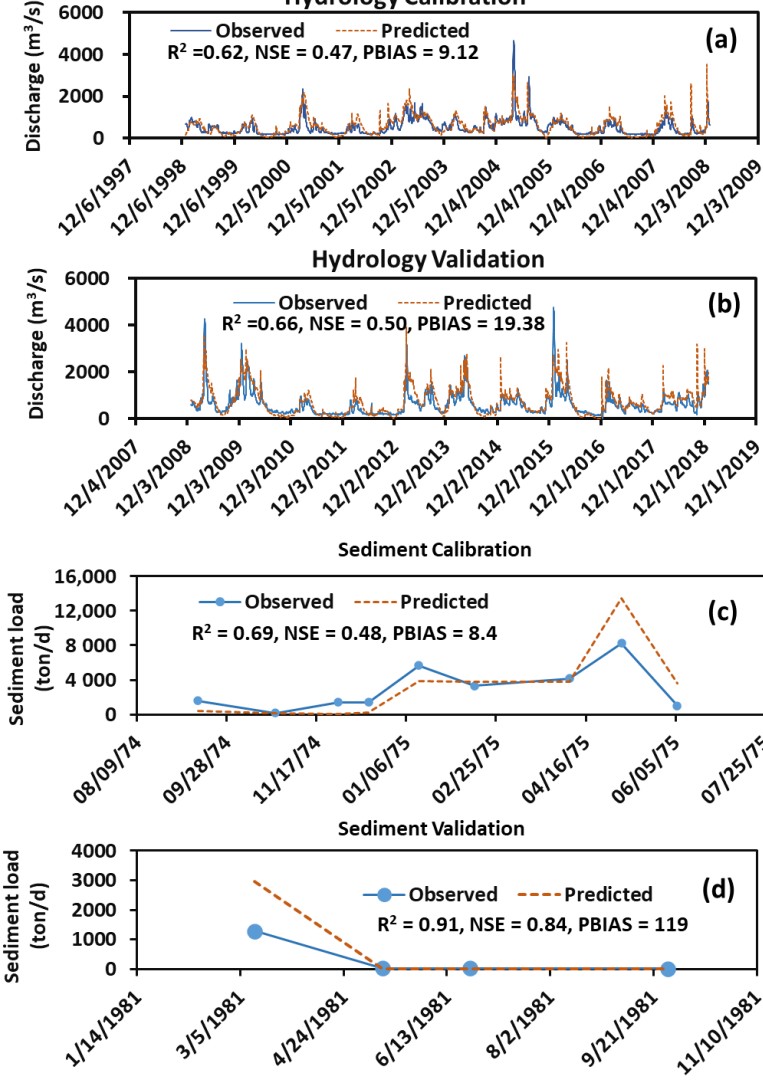

**Figure 3.** Comparisons of the observed and predicted daily discharges (**a**,**b**) and sediment loads (**c**,**d**) during the model calibration and validation.

### 3.2. Daily, Monthly, and Annual Sediment Load

The changes in the daily, monthly, and annual sediment load at the PRB outlet into the NGOM for the past and future scenarios are, respectively, shown in Figures 4 and 5. All the changes are significant at $\alpha$ = 0.01 between the past and future scenarios among the daily, monthly, and annual values based on the Kolmogorov–Smirnov test. In general, the daily sediment load varied from date to date. The average daily sediment loads at the basin outlet were 872 and 753 ton/d, respectively, for the past and future scenarios. In other words, there was a 16% decrease in the sediment load for the future scenario in comparison to that of the past scenario. We attributed the future sediment load reduction to the decrease in the future daily average maximum precipitation, although the average annual precipitations between the past and future scenarios were very close (Table 2). The basin average daily maximum precipitation was 38 mm for the past scenario but was 29 mm for the future scenario. The latter was about 31% less than the former. The lower future daily average maximum precipitation had resulted in the lower annual average surface runoff (Table 2). The lower surface water runoff would result in the lower soil erosion and thus the lower sediment load in the streams. The results suggested that the higher precipitation rate is an important factor for sediment load in the streams.

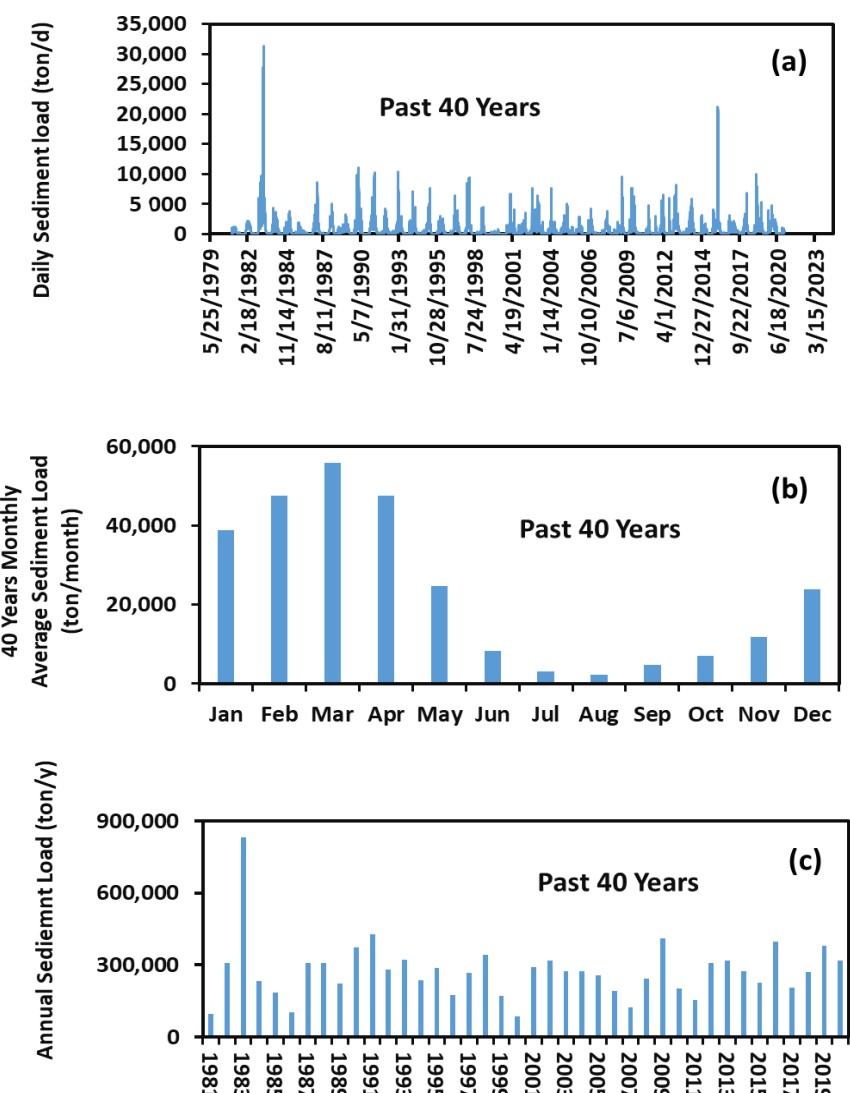

**Figure 4.** Daily (**a**), monthly (**b**), and annual (**c**) sediment loads at the PRB outlet over the past 40 years from 1981 to 2020.

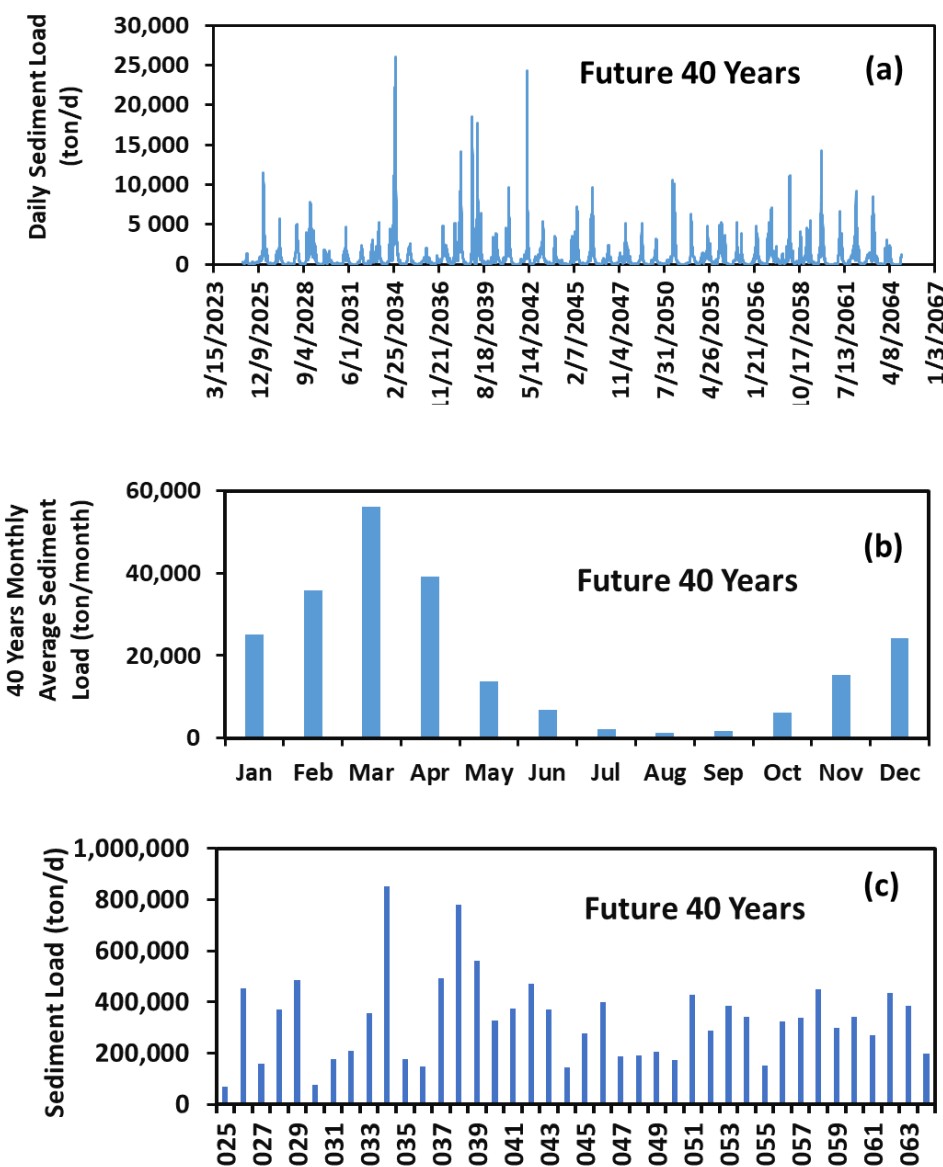

**Figure 5.** Daily (**a**), monthly (**b**), and annual (**c**) sediment loads at the PRB outlet over the future 40 years from 2025 to 2064.

Variations of the 40-year monthly average sediment load in the past scenario are shown in Figure 4b. The maximum was 55,776 ton/month in March, while the minimum was 2257 ton/month in August. Thorne et al. [32] estimated the historical sediment load in the lower Mississippi River and reported that the largest monthly sediment load was in April and the smallest one was in September, at the Tarbert Landing watershed that is adjacent to the PRB, which were somewhat comparable to our findings. Analogous to the case of the past scenario, the same temporal monthly trend was observed for the future scenario (Figure 5b). The results indicated that the monthly variation in sediment load did not change for the past and future scenarios.

A seasonal pattern of the past and future sediment loads can be deducted from Figures 4b and 5b. That is, the seasonal sediment load for both scenarios increased from winter to spring and decreased from spring through summer to fall with the following order: spring > winter > summer > fall. This order was consistent with that reported by Thorne et al. [32]. Among the four seasons, winter and spring accounted for 87% of the sediment load for the past scenario and 86% of the sediment load for the future scenario. Winter and spring are wet seasons while summer and fall are dry seasons in the PRB.

Heavy and frequent precipitation in the wet seasons would produce more sediment load in streams.

**Table 2.** Annual average precipitation, surface water runoff, and sediment load for each watershed within the PRB.

| Watershed Name | HUC Number | Past Annual Average Precipitation (mm) | Future Annual Average Precipitation (mm) | Past Daily Average Maximum Rainfall (mm) | Future Daily Average Maximum Rainfall (mm) | Past Annual Average Surface Water Runoff (mm) | Future Annual Average Surface Water Runoff (mm) |
|---|---|---|---|---|---|---|---|
| Upper Pearl River Watershed | 3180001 | 1459 | 1461 | 36 | 26 | 250 | 178 |
| Middle Pearl River Watershed | 3180002 | 1467 | 1469 | 35 | 28 | 290 | 220 |
| Lower-Middle Pearl River Watershed | 3180003 | 1535 | 1514 | 39 | 29 | 108 | 75 |
| Lower Pearl River Watershed | 3180004 | 1617 | 1629 | 40 | 30 | 242 | 181 |
| Bogue Chitto River Watershed | 3180005 | 1612 | 1608 | 41 | 30 | 270 | 194 |
| **Basin Average** | | 1538 | 1536 | 38 | 29 | 232 | 170 |

Analogous to the case of the daily sediment loads, the annual sediment loads for the past and future scenarios varied from year to year (Figures 4c and 5c). The largest annual sediment loads were found in 1983 for the past scenario (Figure 4c) and in 2034 for the future scenario (Figure 5c). These occurred because of the largest daily sediment loads during those years (Figures 4a and 5a). Overall, the 40-year sediment load was 15,926,342 tons for the past scenario and was 11,010,298 tons for the future scenario. The latter was 30% less than the former. The results indicated a reduction of the sediment load in the next 40 years (from 2025 to 2064) from the PRB into the NGOM due to the reduction of surface water runoff because of a decrease in the average daily maximum precipitation in the next 40 years.

A Mann–Kendall analysis was performed for annual precipitation, discharge, and sediment load over both the past and future 40 years. The Mann–Kendall statistic $\tau$ ranges from −1 to 1 and measures the relationships between variables and times. If $\tau = 0$, no relationship exists, while $\tau = 1$ indicates a perfect increasing trend, and −1 a perfect decreasing trend. The *p*-value is a statistical measure of a trend and, if $p \leq 0.05$, there is a monotonic trend [33]. While there were some decreasing ($-\tau$) and increasing ($+\tau$) trends in some hydrological variables, our Mann–Kendall statistical test revealed no significant trends for the annual precipitation, discharge, and sediment load over both the past and future 40 years (Figure 6).

The plots of annual sediment load with annual discharge showed the following good linear correlations: $Y = 2.53X - 100{,}612$ with $R^2 = 0.701$ for the past 40 years and $Y = 3.24X - 117{,}356$ with $R^2 = 0.84$ for the future 40 years. Similar correlations were obtained for watersheds at various locations in the USA by Ouyang [1].

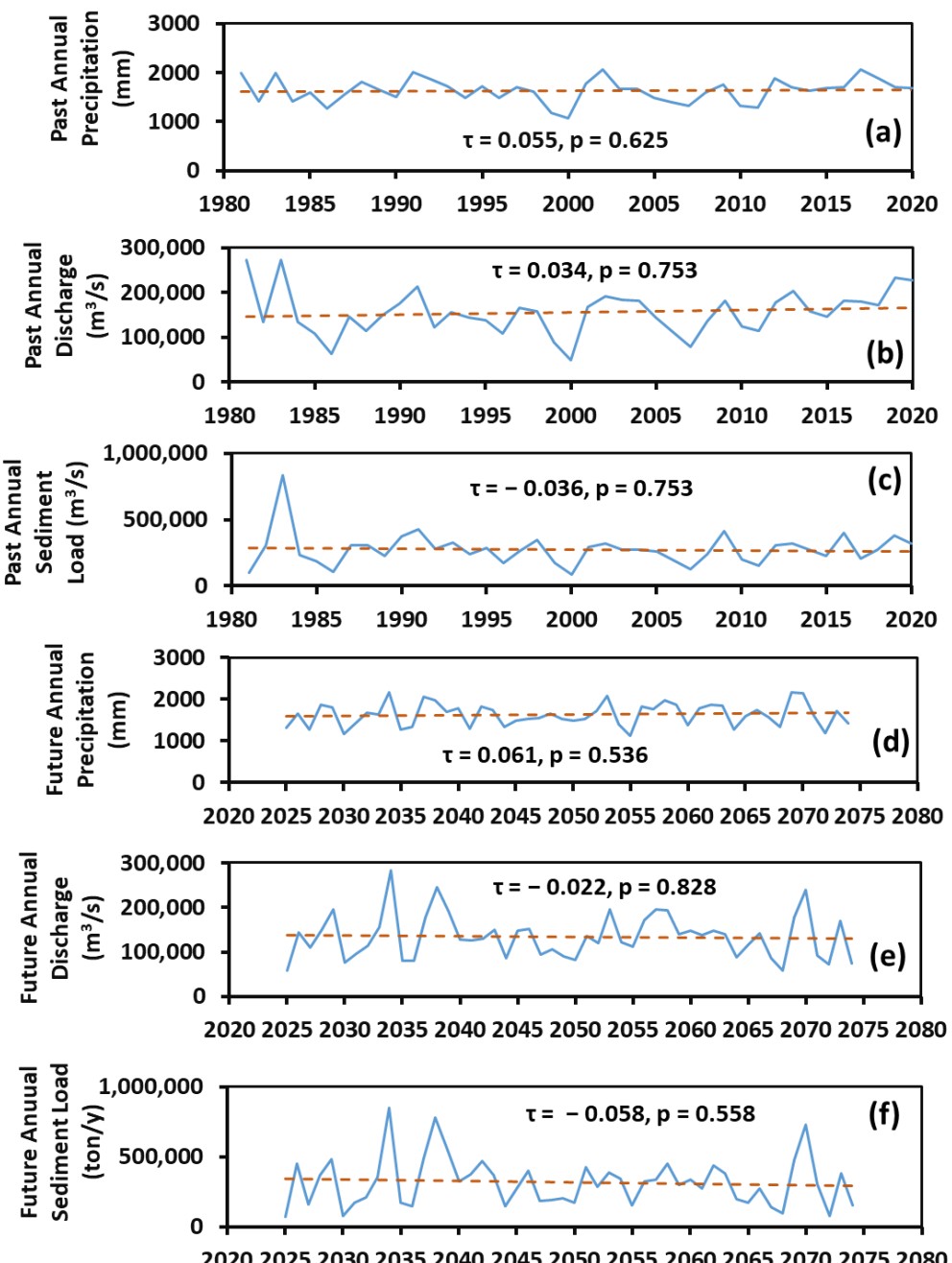

**Figure 6.** Trend analysis (Mann–Kendall) for the past annual precipitation (**a**), discharge (**b**), and sediment load (**c**) as well as for the future annual precipitation (**d**), discharge (**e**), and sediment load (**f**).

### 3.3. Effects of Afforestation under a Future Climate

Afforestation is a field process to grow trees in the non-forest lands to create forest plantations. Afforestation conserves rainwater, diffuses surface runoff, and absorbs pollutants, which mitigates river flooding, reduces stream sediment load, and generates a higher quality water (i.e., clean water) [5]. The differences in the annual average sediment yield between the future and future afforestation scenarios among the five watersheds within the PRB are shown in Figure 7. The differences were calculated by subtracting the values of the future scenario from the values of the future afforestation scenario. In the future afforestation scenario, the rangeland of the middle Pearl River watershed was converted to the mixed forest land.

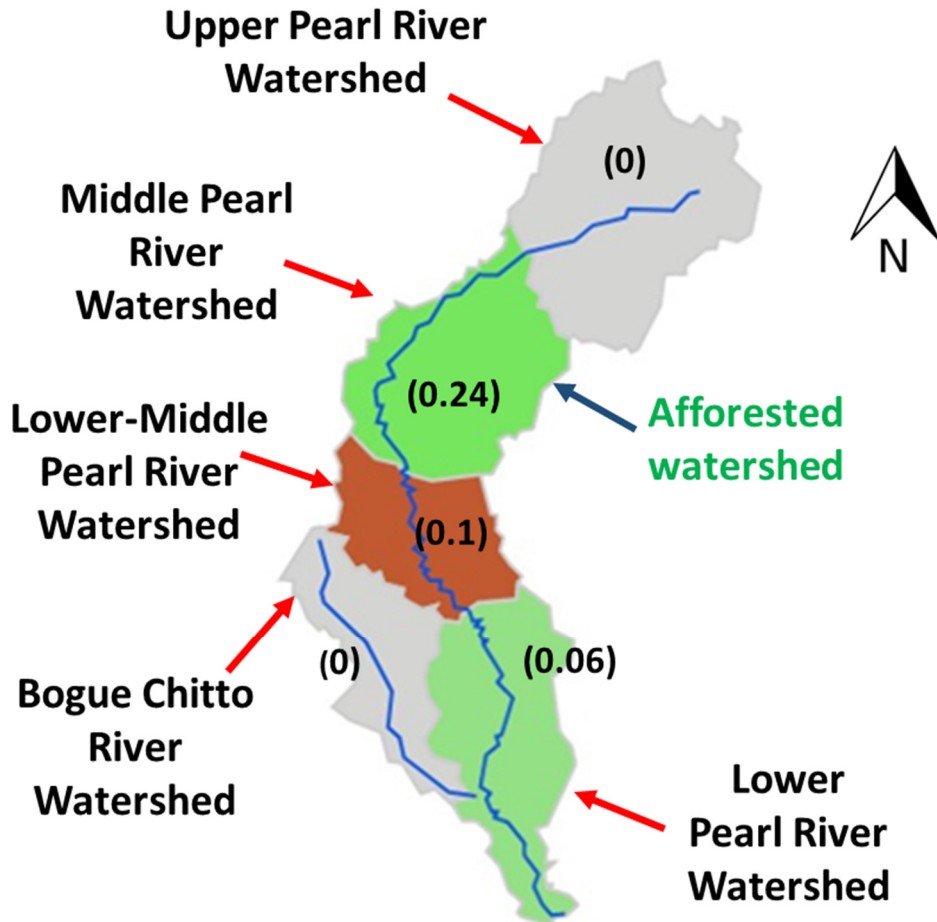

**Figure 7.** Differences (values in paratheses at ton/ha/y) in annual average sediment yield between the future and future afforestation scenarios among the watersheds within the Pearl River basin.

Little to no difference in the annual average sediment yield was observed for the upper Pearl River watershed as afforestation in the middle Pearl River watershed did not affect the hydrological processes and sediment transport at the upstream watershed. No difference in the annual average sediment yield was observed at the Bogue Chitto River watershed either after afforestation because the streams in the middle Pearl River watershed did not flow through the Bogue Chitto River watershed. There was a 0.24 ton/ha/y reduction in the annual average sediment yield at the middle Pearl River watershed after afforestation. This afforested watershed further affected its downstream watersheds, i.e., the lower-middle Pearl River and lower Pearl River watersheds. There were 0.1 and 0.06 ton/ha/y reductions at the lower-middle Pearl River and lower Pearl River watersheds, respectively. Overall, a 10% reduction in the annual average sediment yield with afforestation was observed as compared to without afforestation. The results indicated that afforestation reduced the sediment load at the afforested watershed and its downstream watersheds.

## 4. Summary

The HAWQS-based PRB model was developed to simulate sediment load from the PRB into the NGOM. Good agreements were obtained between model predictions and field measurements during the model calibration. Three simulation scenarios were then created to project sediment loads for the past 40 years (from 1981 to 2020), future 40 years (from 2025 to 2064), and future 40 years with afforestation.

Over the past and future 40 years, the maximum sediment load occurred in April and the minimum sediment load in August. The pattern of the monthly variation in sediment load was the same in both the past and future scenarios.

A seasonal pattern of the past and future sediment loads was observed: increasing from winter to spring and decreasing from spring through summer to fall with the following order: spring > winter > summer > fall. Our study revealed that there was 16% less sediment load from the PRB into the NGOM for the future 40 years than for the past 40 years due to the reduction in surface water runoff that resulted from the decrease in the daily maximum precipitations.

While there were some decreasing ($-\tau$) and increasing ($+\tau$) trends in some hydrological variables, the Mann–Kendall statistical test revealed no significant trends for the annual precipitation, discharge, and sediment load over both the past and future 40 years.

Afforestation reduced the sediment load from the PRB into the NGOM at the afforested watershed and its downstream watersheds. This finding provides useful reference for forest restoration practices.

**Author Contributions:** Conceptualization, Y.O.; methodology, Y.O.; validation, Y.O.; formal analysis, Y.O.; investigation, Y.O.; resources, Y.O.; data curation, Y.O.; writing—original draft preparation, Y.O.; writing—review and editing, Y.O., Y.H., P.B.P., Y.W., J.M.G., P.V.C. and C.T. All authors have read and agreed to the published version of the manuscript.

**Funding:** This research received no external funding.

**Institutional Review Board Statement:** Not applicable.

**Informed Consent Statement:** Not applicable.

**Data Availability Statement:** Data will be available by the authors upon request.

**Acknowledgments:** The views expressed in this article are those of the authors and do not necessarily reflect the views or policies of the United States Department of Agriculture and the U.S. Environmental Protection Agency.

**Conflicts of Interest:** The authors declare no conflict of interest.

## Abbreviations

| | |
|---|---|
| PRB | Pearl River basin |
| HAWQS | Hydrologic and Water Quality System |
| HRU | Hydrologic Response Unit |
| HUC | Hydrologic Unit Code |
| NGOM | Northern Gulf of Mexico |

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
