# Peer review of "Projection of Sediment Loading from Pearl River Basin, Mississippi into Gulf of Mexico under a Future Climate with Afforestation"

_climate, doi:10.3390/cli11050108_

Round 1

Reviewer 1 Report

        These researches required a large volume of work. The article is very well structured. Through this work, it was possible to make a forecast of the devastating effect of waters pollution and loads sedimentation, and human activity is found among the causes that determine this fact.

Author Response

Reviewer #1

  1. These researches required a large volume of work. The article is very well structured. Through this work, it was possible to make a forecast of the devastating effect of waters pollution and loads sedimentation, and human activity is found among the causes that determine this fact.

Response:  We appreciate the reviewer’s time and efforts for evaluating our manuscript.

Reviewer 2 Report

Dear Authors,

I made some observations, especially indicating that the authors improve formatting, clarity and the organization of ideas. Most of them point out that the manuscript has a more aligned approach to the Journal scope (Climate). This point really caught my attention and requires more care. The conception of the manuscript is good but it has some methodological gaps and lacks greater systematization and accuracy of information/findings. I highlighted in the attached file some points that was essential to make my decision: major revisons.

Best regards.

Author Response

Reviewer #2

  1. I made some observations, especially indicating that the authors improve formatting, clarity and the organization of ideas. Most of them point out that the manuscript has a more aligned approach to the Journal scope (Climate). This point really caught my attention and requires more care. The conception of the manuscript is good but it has some methodological gaps and lacks greater systematization and accuracy of information/findings. I highlighted in the attached file some points that was essential to make my decision: major revisions.

Response: We appreciate the reviewer’s time and efforts for evaluating our manuscript. In this revision, we have carefully looked at the texts highlighted by the reviewer in the pdf file and revised manuscript as necessary. All the changes are tracked for the reviewers’ conveniences.

  1. The introductory text presents an appropriate approach to the proposed objectives. However, the novelty, economic impact and practical applicability of this study should be highlighted more.

Response:  The following sentences in Lines 75 to 79 of the revised manuscript addressed this comment: “However, a thorough literature review reveals that very few efforts have undertaken to estimate the contributions of forest watersheds to sediment delivery from the LMRB into the NGOM under a changing climate.  This knowledge is critical to the assessment of relations among afforestation, climatic, environmental, and economic impacts in the region.”

  1. The following comment in Lines 58 and 60:

Response: Authors appreciate the reviewer’s suggestion. However, we would not be able to remove the phrase “Milliman and Meade” and “Bentley” in the sentences. If we remove them, the sentences start with “[11]”, which are not good. The Climate (this journal) and many other journals use the same format as we did in their publications.

  1. Materials and methods section: I suggest the insertion of methodological flowchart. It will make it simpler to the reader! It is missing the description of the procedures for the elaboration of cartographic and graphic products. Describe why you choose the statistical metrics to evaluate the relationship between the variables.

Response: Thanks for your constructive comment. A flowchart (Figure 2) for the modeling procedures is added in this revision.

In the field of hydrology and water quality modeling, the statistical variables such as R2, Nash Sutcliff efficiency (NSE), and percent bias (PBIAS) are well acceptable and commonly used statistical metrics in model calibration and validation.

  1. Figure 1. This map has many cartographic and visual problems. The scale is missing (something very basic for a map). Two maps should be inserted (in one): one locating the study area in the USA; another emphasizing the location of the study area in both states with information about altitude. And another more detailed with information on fluviometric stations and types of land use and occupation.

Response:  The scale is added to Figure 1 in this revision. The figure is redrawn to somewhat reflect the reviewer’s comment although we feel that the original figure for the study location and land uses are clear.

  1. Considering the proposal of this research, I expected the most robust presentation of the climate characteristics of the study area. Therefore, I recommend the insertion of more detailed information about Geomorphology and especially with details of study Climatology. Line 96 to 98: I suggest the insertion of a set of graphics in the climogram format.

Response:  In the field of hydrology and water quality modeling, it is quite common to give a very brief description of the climate for the study site, but not to provide any climographs. Many recent publications in Climate (this journal) used the same approach (i.e., the brief description of climate for a study site) as we did in this study. Overall, our major focus in this study is the sediment load and hydrology. It will generate a lengthy manuscript if we add too many graphics.

  1. Line 128: I suggest changing at the beginning of the text and here. Instead of using this nomenclature (#03180004), name as ID 3 and so on.

Response: It is a convention in the field of hydrology to keep the USGS water monitoring number as #03180004. This will help readers search a USGS monitoring station in Google much easier.

  1. Results and Discussion section: The results could have explored the results of the historical series more density with the application for example of the Mann-Kendall test (to verify changes in the selected variables). It was not clear at any time of this manuscript what is the relationship between climate and flow or sediment production.

Response: Following the reviewer’s comment, we have performed the Mann-Kendall analysis for annual precipitation, discharge, and sediment load over both the past and future 40 years (see Fig. 6 and Lines 314 and 321 of the revised manuscript).

  1. Line 161-163: This has to be removed for the "Materials and Methods" item with the description of the procedures for the elaboration of cartographic and graphic products. Describe why you choose these statistical metrics (with references).

Response: Thanks for the comment. However, we feel that it is appropriate to keep the sentence “The goodness-of-fit during the model calibration was determined with coefficient of determination (R2), Nash Sutcliff efficiency (NSE), and percent bias (PBIAS).” as it was for readers’ conveniences.

As we responded to your Comment #4, in the field of hydrology and water quality modeling, the statistical variables such as R2, Nash Sutcliff efficiency (NSE), and percent bias (PBIAS) are well acceptable and commonly used in model calibration and validation.

  1. Lines 165 and 169: There is inconsistency between the periods of calibration. Why are they so different?

Response: Yes, the periods for discharge and sediment calibrations are different because we do not have the measured data for discharge and sediment at the same period.  By the way, this section has been substantially revised to include model validations.

  1. Summary Section: I considered this item without the necessary development to qualify the research as a whole. A denser analysis of the impacts of the changes detected and projected on the study area is missing.

Response:  In the summary section, we outlined our research findings from this study based on our study objectives. We have also added the Mann-Kendall results to this section in this revision.

Reviewer 3 Report

The manuscript is good and important and deals with the impact of human activity when it is not thoughtful or programmed to destroy and distort nature. It presents the increase in soil erosion as a result of the removal of vegetation cover, and what are the future expectations for this subject and the expected treatments for soil preservation.

Usually, abbreviations are not included in the abstract, but in other paragraphs

It is preferable to explain the topography of the studied area because of its effect on increasing soil erosion

Clarification of the relationship between the river load and the amount of discharge to the river

Knowing the area of the river basin is related to the size of the sediments that reach the river. It is best to mention it

Nature of sedimentary rocks in the river basin

Line 256, What is the nature of the weather this year?

A sentence from lines 259-263, rephrase the sentence because the event is expected and not from the past

Author Response

Reviewer #3

  1. The manuscript is good and important and deals with the impact of human activity when it is not thoughtful or programmed to destroy and distort nature. It presents the increase in soil erosion as a result of the removal of vegetation cover, and what are the future expectations for this subject and the expected treatments for soil preservation.

Response:  We appreciate the reviewer’s time and efforts for evaluating our manuscript.

  1. Usually, abbreviations are not included in the abstract, but in other paragraphs It is preferable to explain the topography of the studied area because of its effect on increasing soil erosion.

Response: Changed as suggested.

  1. Clarification of the relationship between the river load and the amount of discharge to the river

Response:  A paragraph was added in this revision to relate sediment load with stream discharge (see Lines 330 to 333 of the revised manuscript).

  1. Knowing the area of the river basin is related to the size of the sediments that reach the river. It is best to mention it Nature of sedimentary rocks in the river basin

Response: The Pearl River basin is basically the alluvial deposition located in the lower Mississippi River flood plain.

  1. Line 256, What is the nature of the weather this year?

Response:  As shown in the Figure 6 of the revised manuscript, there were high rates of annual precipitation (Fig. 6a) and discharge (Fig. 6b) in 1983.

  1. A sentence from lines 259-263, rephrase the sentence because the event is expected and not from the past.

Response: Rephrased.

Reviewer 4 Report

(1) The retrospective and prospective estimates of sediment load in the basin of the studied river obtained in the manuscript, based only on the calibration for one year of observations (1981), are very alarming. The lack of data for other years on sediment load does not justify such a dubious approach. Moreover, the authors extrapolate the 1981 sediment load calibration along the middle course of the river to the entire river basin, which also has no scientific explanation.

(2) Information about receiving meteorological data is not clear in the manuscript. Data from which weather stations were used? Where and how densely are they located in the basin of the studied river? What is the role of reservoirs located in the river basin in the received calibration for 1981?

(3) There is no clear information about the limitations and uncertainties of the forecasts.

(4) Statistics of predictive trends in water runoff and sediment load are not available in the manuscript.

Thus, the manuscript needs significant revision. I cannot recommend it for publication in its current form.

Author Response

Reviewer #4

  1. The retrospective and prospective estimates of sediment load in the basin of the studied river obtained in the manuscript, based only on the calibration for one year of observations (1981), are very alarming. The lack of data for other years on sediment load does not justify such a dubious approach. Moreover, the authors extrapolate the 1981 sediment load calibration along the middle course of the river to the entire river basin, which also has no scientific explanation.

Response: We understand the reviewer’s concern. In this revision, we have included both the model calibration and validation for stream discharge and sediment load (Fig. 3). We believe this revision addressed the reviewer’s concern.

  1. Information about receiving meteorological data is not clear in the manuscript. Data from which weather stations were used? Where and how densely are they located in the basin of the studied river? What is the role of reservoirs located in the river basin in the received calibration for 1981?

Response: As we stated in Lines 122 to 124 of the revised manuscript (or Lines 117-118 of the original manuscript), the meteorological data were pre-loaded into the HAWQS model for watersheds in the continental USA (https://hawqs.tamu.edu/#/).  In other words, once we decided which watershed to simulate, we can simply go to the HAWQS model and select the watershed of interest, the meteorological data (both the past and future) were already there, which is very time-saving. These data were prepared by the HAWQS model team and rigorously reviewed by US-EPA and SWAT model Teams. That is the reason we use the HAWQS model.

  1. There is no clear information about the limitations and uncertainties of the forecasts.

Response:  The HAWQS model is a customized version of the SWAT model. Any limitations inherited by the SWAT model will also apply to the HAWQS model.

  1. Statistics of predictive trends in water runoff and sediment load are not available in the manuscript. Thus, the manuscript needs significant revision. I cannot recommend it for publication in its current form.

Response: Mann Kendall analysis was performed to analyze the trends of annual precipitation, discharge, and sediment load in this revision (Fig. 6). We hope the reviewer will find the revised manuscript much improved and is suitable for publication

Round 2

Reviewer 2 Report

Dear Authors,

I have identified that you have performed all the corrections and changes requested.

The manuscript presented a high degree of improvement quality and my decision is to "accept". 

Congratulations!

Kind regards.

Author Response

Reviewer #2

Dear Authors,

I have identified that you have performed all the corrections and changes requested.

The manuscript presented a high degree of improvement quality and my decision is to "accept".

Congratulations!

Kind regards.

Response: We appreciate the reviewer’s time and effort to evaluate our manuscript again.

Reviewer 4 Report

In addition to the limitations inherent in the model used (the HAWQS model), there are always so-called "external" limitations that are not related to it. Did you have them? Information about them will allow the reader to be more aware of the weaknesses of your research. Be critical of your conclusions.

Author Response

Reviewer #4

In addition to the limitations inherent in the model used (the HAWQS model), there are always so-called "external" limitations that are not related to it. Did you have them? Information about them will allow the reader to be more aware of the weaknesses of your research. Be critical of your conclusions.

Response:  Thanks for the comment.  The following sentences have been added to the revised manuscript (see Lines 129 to 131): “However, it should be noted that any limitations inherited by the SWAT model are also applied to the HAWQS model.  Additionally, HAWQS has fewer input parameters that can be used for model calibration and is not as flexible as SWAT.” 
